# Phytochemical Composition and Antioxidant Activity of *Passiflora* spp. Germplasm Grown in Ecuador

**DOI:** 10.3390/plants11030328

**Published:** 2022-01-26

**Authors:** William Viera, Takashi Shinohara, Iván Samaniego, Atsushi Sanada, Naoki Terada, Lenin Ron, Alfonso Suárez-Tapia, Kaihei Koshio

**Affiliations:** 1Faculty of International Agriculture and Food Studies, Tokyo University of Agriculture, Sakura gaoka 1-1-1, Setagaya, Tokyo 156-8502, Japan or william.viera@iniap.gob.ec (W.V.); t3shinoh@nodai.ac.jp (T.S.); a3sanada@nodai.ac.jp (A.S.); nt204361@nodai.ac.jp (N.T.); koshio@nodai.ac.jp (K.K.); 2Fruit Program, Tumbaco Experimental Farm, National Institute of Agricultural Research (INIAP), Av. Interoaceánica km 15 and Eloy Alfaro, Tumbaco 170902, Ecuador; ivan.samaniego@iniap.gob.ec; 3Zoonosis International Center, Universidad Central del Ecuador (UCE), Quito 170521, Ecuador; ljron@uce.ude.ec; 4Graduate School of Agroindustry and Food Science, Universidad de las Américas (UDLA), Quito 170503, Ecuador

**Keywords:** phytochemical composition, antioxidant activity, *Passiflora* spp. germplasm, Ecuador, polyphenols, organic acids

## Abstract

Tropical fruits are in high demand for their flavor and for their functional composition because these compounds are considered nutraceuticals. Passion fruit production is of economic importance to Ecuador; however, several *Passiflora* species are grown and each has to be analyzed to identify their phytochemical composition. In this study, the polyphenol, flavonoid, carotenoid, vitamin C, sugar and organic acid contents were determined. Six different *Passiflora* spp. germplasms were analyzed, coming from *Passiflora edulis* f. *flavicarpa*, *Passiflora alata*, *Passiflora edulis* f. *edulis* and unidentified *Passiflora* species (local germplasm). Measurement techniques included reflectometry for vitamin C, spectrophotometry for antioxidant compounds and HPLC for sugars and organic acids. Data were analyzed by principal component analysis, correlation and analysis of variance. Results showed that INIAP 2009 and P10 showed a high amount of polyphenols, antioxidant activity and citric content. Sweet passion fruit had the lowest vitamin C content while Gulupa showed the highest content. In terms of the local germplasm, POR1 showed the lowest content of flavonoids while PICH1 had high flavonoid and carotenoid content. Polyphenols were the main compounds that influenced antioxidant activity. This phytochemical information adds value to passion fruit as a nutraceutical source.

## 1. Introduction

The species of the *Passifloraceae* family are grown in tropical and subtropical regions worldwide [1]. In South America, the main countries where passion fruit is grown are Brazil, Ecuador, Peru and Colombia [2]. This fruit is consumed fresh as juice (locally) because around 40% of the fruit is composed of pulp, or it is marketed (world-wide) as a concentrate and is desirable as a beverage component due to its nutritional characteristics, color, acid taste and exotic aroma [3,4,5].

There are several species of *Passifloras* grown in South América which are phenotypically different [6], mainly in terms of fruit size and peel color. Moreover, other wild species are used, and some may intercross with the domesticated and semidomesticated species (local germplasm) [7]. *Passiflora edulis* f. *flavicarpa* (yellow passion fruit) is the most cultivated species in Ecuador, Brazil and Peru; while *P. edulis* f. *edulis* is mostly grown in Colombia [8]. In Ecuador, yellow passion fruit is grown in the coastal lowlands (Manabí, Quevedo, Guayas) while purple passion fruit is cultivated in the highlands (Imbabura). *P. edulis* f. *flavicarpa* reaches fruit measurements of 6 to 7 cm in diameter and 6 to 12 cm in length [9], fruit weight of 174 g, and peel thickness of 7.4 mm [10]. *P. edulis* f. *edulis* (purple passion fruit), mainly grown in Colombia, is almost round with a diameter of about 5 cm, thinner peel than yellow passion fruit, and weight between 42 and 68 g [8]. *P. alata* (Sweet passion fruit) is mainly grown in Brazil; it has a fruit weight between 192 and 243 g, length of 9.6 cm, and diameter of 7.1 cm [11,12]. In Ecuador, there are some local cultivars called “Criollo” (*Passiflora* sp.) which have not been botanically described but are grown by farmers; the fruit are bigger or similar in size to those of *P. edulis* f. *edulis* but smaller than *P. edulis* f. *flavicarpa*, reaching around 93 g of fruit weight and 5 mm of peel thickness [10].

Consumption of fruits is clearly associated with health benefits such as improving the immune system, reduction of cellular oxidative damage and protection against cancer development [13]. These properties are attributed to the presence of phytochemicals and nutrients with antioxidant properties; therefore, fruit antioxidant compounds have been reported to be beneficial for the human diet [14]. In addition, there is a great interest on the use of natural antioxidants in food products because of studies that show possible adverse effects due to the consumption of synthetic antioxidants [15]. Nevertheless, the concentration of these compounds is strongly influenced by climate and soil conditions, crop developmental stage and genetic factors [16].

Natural biophenols are a broad group of molecules found only in plants, and these molecules possess an extraordinary antioxidant power, being produced as secondary metabolites by the plant for protection against attack by bacteria, fungi, and insects [17]. Phenolic compounds are the major group of natural antioxidants (biocompounds) because of their great diversity and broad distribution, contributing greatly to the total antioxidant activity [18]. Among the main antioxidant compounds are polyphenols, flavonoids and carotenoids [19]. Polyphenols and carotenoids are biomolecules related to fruit color [13]. 

*Passiflora* species have been found a good source of antioxidant compounds, differing in type according to the species [1]. *P. edulis* has been associated with higher values of total phenolic compounds, vitamin C, total carotenoids, and antioxidant activity (ABTS) [20]. Consequently, it has been stated that passion fruit pulp is a good source of secondary metabolites such as phenols, carotenoids, flavonoids and tannins but also of other bioactive compounds such as saponins [1,21,22]. In addition, this fruit is a source of vitamin C, sugars and organic acids [5,23,24].

Polyphenols from passion fruit have been reported to have potential beneficial effects in many pathological conditions [25]; hence, this fruit is a good source of these valuable biocompounds [26]. Catechin is the main flavonoid found in *P. edulis* and this species contains more of this biocompound than other *Passiflora* species [20]. In terms of carotenoids, beta carotene is naturally abundant in yellow orange fruits [27], being also reported as the main carotenoid in yellow, purple and orange passion fruit [1,18,28]. Vitamin C is found in many fruits [14] such as passion fruit [5], especially when they are grown under organic as compared to conventional systems [18]. Barbosa et al. have reported low content of citirc acid in yellow passion fruit [20], whereas Oliveira-Folador reported low concentrations of ascorbic acid in the same cultivar [29].

Studies have mainly determined the antioxidant activity of the yellow passion fruit (*Passiflora edulis* f. *flavicarpa*) in order to use the pulp of this fruit as an input for food products [1,28]. Barbosa de Oliveira found that the total antioxidant activity is higher in yellow passion fruit produced under conventional management [18], and it has been reported to have an antioxidant activity of 33% in this fruit [28].

In fruit, sugar is the main factor influencing fruit quality [30]. Mamede et al. found that the pulp of Sweet passion fruit (*P. alata*) has a high content of sugar; this fruit is sweeter and less acidic compared to traditional commercial passion fruits [31]. In addition, yellow and purple passion fruit have more sugar content than other more sour species [23]. The main sugars present in the passion pulp are glucose, fructose and sucrose [24]. Matute et al. observed similar amounts of glucose and fructose in the pulp of yellow passion fruit [21] but Barbosa et al. found different values of these two sugars [20]; sucrose is in a lesser proportion in passion fruit [23]. The main organic acids present in yellow passion fruit pulp are citric and malic acids [20]. The content of these organic acids in yellow and purple passion fruit is variable, depending on the fruit developmental stage and environmental conditions [24,32].

In Ecuador, studies of yellow passion fruit accessions have been carried out for basic fruit quality variables [10,33]; however, there is almost no information on phytochemical compounds of the different *Passiflora* germplasm grown in the country. Currently, the phytochemical composition of the fruits is of great importance to highlight the benefit of their consumption for human health; thus, studies have been carried out mainly on passion fruit commercial cultivars (*P. edulis* f. *flavicarpa* and f. *edulis*); however, this study also considers local cultivars that are little exploited but can be a potential source of biocompounds. In addition, *P. alata* was also considered; it is an underutilized species and there is little information on the chemical composition of its fruits but it is appreciated for its medicinal properties [34]. Consequently, the objective of this study was to determine the content of antioxidant compounds, sugars and organic acids in different passion fruit germplasm grown in Ecuadorian environmental conditions, adding value to this fruit as a nutraceutical source and generating information that may be used for further breeding programs.

## 2. Materials and Methods

### 2.1. Experimental Site and Vegetal Material

The research was carried out at the Nutrition and Quality Laboratory (ISO/IEC 17025) of the National Institute of Agricultural Research (INIAP), located in Cutuglahua (Pichincha province), 00°22′57″ South and 78°33′18″ West.

Fruit of the different passion fruit germplasms were harvested from research sites belonging to INIAP (Table 1) and were harvested in maturity grade 5 (100% color change on the tree) [35,36] from 1 year-old plants.

### 2.2. Chemical Reagents

A Milli-Q Academic water purification system (Millipore, Sao Paulo, Brazil) was used to obtain deionized water. The standards of (+) catechin, gallic acid, cyanidin-3-glucoside chloride, ABTS (2,2-azinobis-3-ethyl-benzothiazoline-6-sulfonic acid), and Trolox (6-hydroxy-2,5,7,8-tetramethylchroman-2-carboxylic acid) were obtained from Sigma Aldrich (St. Louis, MO, USA). The analytical grade solvents and reagents were acquired from Merck (Darmstadt, Germany). 

### 2.3. Preparation of the Samples

Pulp was extracted from the fruit and dried by lyophilization. Dried samples were ground in a Retsch model ZM 200 mill (Hann, Germany) then passed through a stainless-steel sieve (1 mm mesh) to a uniform particle size. 

### 2.4. Color Assessment

Two hundred grams of lyophilized fruit pulp of each germplasm was homogenized in a blender. Thirty grams of each sample was placed in a Petri dish. The dishes were placed on a white surface and divided into four equal parts. Measurements were carried out in triplicate in each quarter and at the center of the plate [13]. Color was measured using a ColorTec-PCM handheld colorimeter (ColorTec, Clinton, NJ, USA), with a measurement angle of 10°, Illuminator D65, and aperture of 8 mm. The chromatic properties were defined by the L* a* b* color method of the CIE (Commission Internationale de l’Eclairage) and were expressed as L* (lightness), a* (red/green), and b* (blue/yellow) coordinates. Hue angle (°H) and Chroma (C*) were calculated with the a* and b* values with the following formulae: °H = tan^−1^ (b/a) and C* = (a^2^ + b^2^)^1/2^

### 2.5. Preparation of Extracts

Phytocomponent extraction was carried out by the method of Hue et al. [37]. Dry sample (0.3 g) was placed in 15 mL plastic centrifuge tubes and 5 mL of methanol/water/formic acid solution added (70:30:0.1 *v*/*v*/*v*). A shaking extraction process, using FAST PREP 24 (MP Biomedicals, Fisher Scientific, Hampton, VA, USA) was followed for 5 min and then the sample was placed in an ultrasound bath (Cole-Palmer, Chicago, IL, USA) for 10 min. It was centrifuged in a 4-16KS centrifuge (Sigma, Neustadt, Germany) for 10 min at 5500 rpm (2706× *g*). The supernatant was separated and transferred to a 25 mL amber volumetric balloon. This process was repeated three times and brought to a volume of 25 mL with the extraction solution. This extract was also used to estimate antioxidant activity (AA).

For the analysis of organic acids, total and reducing sugars, 1 g of dry sample was weighed and 70 mL of water type I was added, stirred for 30 min and centrifuged for 10 min in a Sigma 4-16KS centrifuge; the supernatant was transferred to a 100 mL volumetric balloon and graded with water type I.

### 2.6. Vitamin C

Vitamin C (ascorbic acid) was measured using a reflectometer (RQflex plus 10, Merck, Germany). First, 30 g of pulp was weighed and made up to 200 mL with distilled water. A 20 mL aliquot was taken and tested by immersing an ascorbic acid strip in the reflectometer. The final result was expressed in mg 100 g pulp^−1^ (fresh weigh basis), according to the following formula:$$\mathrm{Vitamin}\;C=\frac{L\;\times \;V}{\mathrm{Sw}}$$
where L = reflectometer lecture (mg/L^−1^), V = final volume (mL) and Sw = sample weight (g).

### 2.7. Total Polyphenol Content Quantification

Total polyphenol content quantification was carried out by UV-visible spectrophotometry [38]. The diluted extract (1 mL) was placed in a 15 mL test tube and 6 mL of distilled water and 1 mL of Folin–Ciocalteau reagent added, then the mixture was rested for 3 min. After that, 2 mL of 20% Na_2_CO_3_ (*w*/*v*) was added and heated to 40 °C for 2 min. The absorbance of the blue chromophore was measured at 760 nm using a UV-VIS spectrophotometer, model 2600 (Shimadzu, Kyoto, Japan). Five extraction cycles were necessary to obtain total polyphenol content recovery. Quantification was carried out by the interpolation of the corresponding absorbance value of each sample on a calibration curve made with gallic acid at 0–100 mg gallic acid/L. Results were expressed as mg of gallic acid equivalents per gram of dry sample (mg GAE g^−1^ DW).

### 2.8. Total Flavonoid Content

Total flavonoid content was measured using a UV-visible spectrophotometer by the method of Zhishen et al. [39]. The diluted extract (1 mL) was placed in a 15 mL tube with 4 mL of distilled water and the mixture homogenized. Next, 0.3 mL of 5% sodium nitrite (*w*/*v*) and 0.3 mL of 10% aluminum chloride (*w*/*v*) were added; the sample was left to rest for 5 min after the addition of each reagent. Finally, 2 mL of 1N NaOH was added to distilled water to a volume of 10 mL. The absorbance was measured in the pink chromophore at 490 nm using a UV-VIS spectrophotometer, model 2600 (Shimadzu, Kyoto, Japan). Five extraction cycles were required to recover 100% of the flavonoid content. The quantification was performed by the interpolation of the corresponding absorbance value of each sample on a calibration curve made with (+)—catechin, 0–100 mg (+)—catechin/L. Results were expressed as mg of catechin equivalents per gram of dry sample (mg catechin g^−1^ DW).

### 2.9. Total Carotenoid Content

Total carotenoid content (TCC) was measured following the method described by Llerena et al. [13], in the absence of light and oxygen and using 1.0 g of the freeze-dried sample. The extraction was done using 50 mL of a solvent mixture composed by hexane 50%, ethanol 25%, acetone 25% (*v*/*v*/*v*), 0.1% of butylated hydroxytoluene (BHT) (p/v) and 5 g of calcium chloride (p/v). These elements were added gradually, one at a time. The mixture was mixed for 20 min in a refrigerated water bath at 4 °C. The separation phase was achieved by adding 15 mL of distilled water over 10 min. The extract was filtered and transferred to a separating funnel. The organic phase was transferred to a volumetric flask. Hexane was added to a total volume of 50 mL. The determination of total carotenoid content was made using the UV-VIS spectrophotometer, model 2200 (Shimadzu, Kioto, Japan) at 450 nm. Results were expressed as µg of β carotene g^−1^ dry weight of pulp. TCC was calculated based on the following equation:$$\mathrm{TCC}=\frac{A\;\times \;\mathrm{VT}\;\times 104}{2592\times \;W}$$
where A is the absorbance at 450 nm, VT is the total volume, 104 is the conversion constant expressed as µg g^−1^, 2592 is the molar extinction coefficient of β carotene in hexane and W is the weight of the sample.

### 2.10. Antioxidant Activity as Measured by the ABTS Method

Antioxidant activity (AA) was estimated by the 2,2-azinobis (3-ethyl-benzothiazoline-6-sulfonic acid) cation bleaching method (ABTS•+) [38]. The ABTS•+ solution (7 mM) and the potassium persulfate solution (2.45 mM) were mixed in a 1:1 ratio (*v*/*v*). The absorbance of the prepared ABTS•+ solution was measured the following day, and was diluted using a phosphate buffer until getting an absorbance of 1.1 ± 0.01 at 734 nm. In 15 mL test tubes containing 200 µL of sample, 3.8 mL of the ABTS•+ working solution was added and the solution rested for 45 min. Absorbance was measured at 734 nm by a UV-VIS spectrophotometer, model 2600 (Shimadzu, Kyoto, Japan). The AA was estimated by interpolating the absorbance on a calibration curve developed previously with a Trolox standard (0–800 µmol Trolox L^−1^). Results were reported as µmol equivalent Trolox per gram of dry sample (µmol TE g^−1^ of sample DW).

### 2.11. Antioxidant Activity as Measured by the Ferric Reducing Power (FRAP) Method

Ferric reducing power (FRAP) was also measured to estimate the AA [38]. The diluted extract (1 mL) was placed in a 15 mL test tube and 2.5 mL of phosphate buffer at pH 6.6 and 2.5 mL of a 1.0% potassium ferrocyanide solution were added. The mixture was shaken and incubated at 50 °C for 20 min, then 2.5 mL of 10% trichloroacetic acid with 2.5 mL of water and 0.5 mL of 1% FeCl_3_ were added. The mixture was homogenized in a vortex (Mistral Multi-Mixer, Melrose Park, IL, USA). The solution was rested for 30 min in the dark and a green complex (ferrous chloride–potassium ferrocyanide) was formed. The absorbance was measured at 700 nm using a UV-VIS spectrophotometer, model 2600 (Shimadzu, Kyoto, Japan). The AA and results were estimated as for the ABTS method. 

### 2.12. Sugar Content

Total sugar content was determined by the method of Dubois et al. [40]. An aliquot of 1.25 mL of extract was placed in a 15 mL test tube, 2.5 mL of anthrone reagent was added and the tubes were heated in a boiling water bath for 10 min. Then tubes were cooled in a cold water bath for 10 min. The reaction formed a green chromophore and the absorbance at 625 nm was determined. Quantification of total sugars was carried out by interpolation of the absorbance of the samples on a calibration curve made with D − (+) − glucose from 0 to 50 mg of glucose L^−1^. Results were expressed as glucose equivalents in g 100 g^−1^ of dry sample.

In addition, reducing sugar content was determined by the method of Miller [41]. An aliquot of 0.5 mL of the extract was placed in a 15 mL test tube, 0.5 mL of reagent of 1 M NaOH and 1.5 mL of DNS reagent (3,5 dinitrosalicylic acid) were added and the tubes were heated in a boiling water bath for 5 min. The tubes were then cooled in a cold water bath for 10 min and 9.5 mL of water type I was added. The reaction formed a brown chromophore and absorbance at 540 nm was determined. The quantification of reducing sugars was done by interpolation of the absorbance of the samples on a calibration curve performed with D (+) glucose from 0 to 2000 mg of glucose L^−1^. Results were expressed as glucose equivalents in g 100 g^−1^ of dry sample. Finally, the content of non-reducing sugars was calculated as the difference between total sugars and reducing sugars.

### 2.13. Organic Acids Quantification

The contents of citric and malic acids were determined by the method of Espín and Samaniego [42]. An aliquot of the extract was passed through a 0.22 µm PVDF Millipore membrane and placed in a capped amber vial and analyzed by High Performance Liquid Chromatography (HPLC). The analysis was carried out using the Agilent 1100/1200 series (Waldbronn, Germany). The separation was carried out using an Agilent column (300 × 6.5 mm) at a flow of 0.7 mL min^−1^ and at a temperature of 40 °C, using a solution of 0.01 N sulfuric acid in water type I as the mobile phase. For the analysis, 10 µL of the purified extract was injected into the equipment and the organic acids were monitored using a UV-visible detector at 210 nm. The identification and quantification of citric and malic acid was carried out by comparison with their respective standards. Results were expressed in g of each organic acid per 100 g^−1^ of dry sample.

### 2.14. Statistical Analysis

In this research, a completely randomized design was applied, and the experimental unit was a 20 g sample of freeze-dried pulp. In terms of the univariate analysis, the Levenne test was calculated to set the homogeneity of variances and an analysis of variance (ANOVA) was carried out with the data. Tukey test at 5% was used to determine differences among means. Data analysis was carried out in the R statistical program version 4.04 [43].

Pearson correlation coefficients (d.f. = 16) were calculated to measure the linear correlation between two variables; while partial correlations (d.f. = 14) were calculated to measure the degree of association between two random variables, removing the effect of a set of controlling random variables [44]. 

Turning to the multivariate analysis, principal component analysis (PCA) and k-means method were used to analyze data and compare their results. PCA was used to visualize the relationship among variables, and their association with the *Passiflora* species; while the k-means analysis was used for grouping because it is more accurate since it looks randomly for similar individuals to finally form fixed groups based on the best number of clusters obtained by the Elbow method [45]. Consequently, the information obtained from these two methods was complementary. 

## 3. Results

### 3.1. Color Assessment of the Passion Fruit Pulp

Color coordinate values, chromaticity and Hue angle are shown in Table 2 and Table 3.

### 3.2. Analysis of Variance (ANOVA)

#### 3.2.1. Antioxidant Activity and Compounds

Antioxidant compounds and contents of the different passion fruit germplasm and species are given in Table 4. INIAP 2009, P10 and Sweet passion fruit had the highest polyphenol content with 2.09, 2.22 and 2.08 mg g^−1^ (gallic acid), respectively. All passion fruit germplasm, except for Criollo POR1, had flavonoid content of over 0.40 mg g^−1^ (catechin). Criollo PICH1 had the highest carotenoid content (67.73 µg g^−1^), while Sweet passion fruit had the lowest (22.80 µg g^−1^). Gulupa had the highest vitamin C content (30.44 mg 100 g pulp^−1^), whereas Sweet passion fruit had the lowest (6.07 mg 100 g pulp^−1^).

Antioxidant activity of passion fruit germplasm and species are showed in Table 5. In terms of ABTS, INIAP 2009 had the highest antioxidant activity (43.00 µmol TE g^−1^), while Criollo PICH1 had the lowest (30.32 µmol TE g^−1^). Turning to FRAP, INIAP 2009 and P10 had the highest antioxidant activity (53.61 and 50.71 µmol TE g^−1^, respectively), while Criollo PICH1 had the lowest (32.14 µmol TE g^−1^).

#### 3.2.2. Sugar and Organic Acid Analysis

Passion fruit germplasm and species differed in sugar composition (Table 5). Sweet passion fruit and Criollo PICH1 had the highest total sugar content (52.47 and 51.47 g 100 g^−1^, respectively), whereas Criollo POR1 had the lowest (41.01 g 100 g^−1^). Sweet passion fruit had the highest reducing sugar content (51.03 g 100 g^−1^), while Gulupa the lowest (40.25 g 100 g^−1^). In terms of non-reducing sugars, INIAP 2009 and P10 had the highest contents (6.57 and 9.90 g 100 g^−1^, respectively), while Sweet passion fruit had the lowest (1.43 g 100 g^−1^).

Table 5 shows the organic acid contents of passion fruit germplasm and species. P10 had the highest citric acid content (31.77 g 100 g^−1^), while Gulupa and Criollo PICH1 had the lowest (12.29 and 12.03 g 100 g^−1^, respectively). In terms of malic acid, P10 had the highest content (5.19 g 100 g^−1^), whereas Criollo POR1 and Criollo PICH1 had the lowest (4.05 and 4.16 g 100 g^−1^, respectively).

### 3.3. Correlation Analysis

A matrix correlation was estimated for the common and partial correlations between antioxidant compounds, sugar and organic acid content (Table 6). The main positive Pearson correlations were between polyphenol and antioxidant activity (ABTS and FRAP), and vitamin C and carotenoid content. On the other hand, there was a high negative correlation between polyphenol and carotenoid contents.

For partial correlations, there was a positive correlation of polyphenol content and antioxidant activity (FRAP), and of vitamin C with carotenoids and malic acid contents.

### 3.4. K-Means Analysis

The Elbow method defined five groups (Figure 1). The first group was formed by individuals of *P. edulis* f. *flavicarpa* (P10 and INIAP 2009), which showed high contents of polyphenol, flavonoid, non-reducing sugars and citric acid, and high antioxidant activity (ABTS and FRAP), while lower contents of carotenoids, total sugars and reducing sugars were detected. The second group consisted of *P. edulis* f. *edulis* (Gulupa) germplasm, which showed high contents of carotenoids, vitamin C and malic acid, and lower contents of non-reducing sugars and citric acid. The third group was composed of *Passiflora* sp. (Criollo POR1), which showed high contents of vitamin C, non-reducing sugars and citric acid contents, and lower contents of flavonoids, total sugars, reducing sugars and malic acid. The fourth group contained *P. alata* germplasm, which showed high contents of polyphenols, total sugars, reducing sugars and malic acid, but low contents of carotenoids, vitamin C, non-reducing sugars and citric acid. Finally, the fifth group was formed by *Passiflora* sp. (Criollo PICH1), which showed high contents of flavonoids, carotenoids, total sugars and reducing sugars; in contrast, they had low contents of polyphenols, antioxidant activity (ABTS and FRAP) non-reducing sugars and citric and malic acid. 

### 3.5. Principal Component Analysis (PCA)

The PCA analysis showed that the first two components explained 74% (Figure 2) of the variance observed in the data. The first component was a contrast between polyphenols vs. carotenoids, while the second component was between antioxidant activity vs. total sugars; in the former component malic acid content also had some influence while in the latter citric acid has some influence.

INIAP 2009 was associated with antioxidant activity and citric acid; P10 with polyphenol and antioxidant activity; Sweet passion fruit with polyphenol content; Gulupa with flavonoids, carotenoids and vitamin C; POR1 with carotenoids and vitamin C; and PICH1 with flavonoids and carotenoids (Figure 2). 

## 4. Discussion

Antioxidants are present in many fruits and vegetables, their unique chemical structure imprinting both their properties and their dietary and therapeutic actions as the literature confirms [46]. However, numerous assessments must be made before they are made directly available for human consumption; they are usually part of a supplementary diet to encourage good health and disease prevention [14]. They are compounds such as polyphenols, flavonoids, carotenoids and vitamin C, present in plants [19]; nevertheless, genotype, geographic area, crop age, maturity and storage conditions can influence the content of these bioactive compounds in fruit [47]. The importance of climatic factors and anthropogenic aspects must not be neglected, as their influence on fruit production is overwhelming [48,49]. In addition, quantitative information about the content of sugars, organic acids and phenolic compounds is relevant to verify the quality index for fruits [50].

### 4.1. Pulp Color

Color traits of fruits depend on factors such as species, soil conditions and harvest period [1]. They are related to the state of maturity, due to the process of accumulation of pigments and the sugar and organic acid contents in the fruits [51]. There was variability among the pulp colors of the different germplasms evaluated in this study; colors ranged from pale yellow (*P. alata*) to orange (the other germplasm) due to the amount of carotenoids, especially β carotene which causes orange color. The pale yellow color is related to phytofuluene [52]. 

Ramos et al. reported values of 8.44 (a*), 42.83 (b*) and 58.5 (L*) for pulp of *P. edulis* f. *flavicarpa*, and values of 0.29 (a*), 17.18 (b*) and 77.65 (L*) for *P. edulis* f. *edulis* [1], showing some differences from those found in our study. The same author observed that the highest brightness (L*) was showed by Gulupa as in this study; however, Sweet passion fruit also got a high value in this parameter. Criollo PICH1 showed the highest value for the parameter a* that indicates the intensity of colors red and green, while P10 and Criollo PICH1 had the highest values for b* which indicated the intensity of yellow and blue.

### 4.2. Univariate Analysis

#### 4.2.1. Polyphenols, Flavonoids, Carotenoids, Vitamin C and Antioxidant Activity

Polyphenol content should be considered a key nutritional and commercial factor [53]. Fruits are the main nutritional sources of polyphenols [54]. The latter are compounds derived from vegetal secondary metabolism. Their effect is recognized in the control they exert against free radicals and reactive oxygen species [53] because they show resistance to oxidation (modulation of oxidative stress, specifically in the activation of transcription) and regulate different types of oxidases in the body [55]. These compounds decrease the risk of chronic, neurodegenerative and cardiovascular disease, and certain cancers [56]. In passion fruit, polyphenol compounds have strong antioxidant activity and its consumption is good for human health [57].

All passion fruit germplasm had higher polyphenol content (>1.40 mg gallic acid g^−1^) than the 0.04, 0.07 and 0.40 mg gallic acid g^−1^ that was reported in yellow passion fruit (*P. edulis* f. *flavicarpa*) in other studies [20,26,58]. However, Silva et al. reported 7.6 mg gallic acid g^−1^ in *P. edulis* f. *flavicarpa* [59], which is higher than found in this study, but Septembre-Malaterre reported a closer value (2.86 mg gallic acid g^−1^) [60]. In terms of purple passion fruit (*P. edulis* f. *edulis*), the result of this study was higher than that reported by Carmona-Hernández et al. (0.47 mg gallic acid g^−1^) [26].

Variety INIAP 2009 and the breeding germplasm P10 showed high content of total polyphenols. Both are yellow passion fruit which is the most grown cultivar in Ecuador and they may be considered as parental for further breeding process because polyphenols are related to high antioxidant capacity. Nonetheless, Sweet pasion fruit (*P. alata*) also showed high content of these compounds; hence this underutilized germplasm can also be taken into account for interspecific crosses or to be used in the food industry.

Flavonoids have antioxidant, pharmacological, anti-inflammatory, antiallergic, antiviral, anticarcinogenic, and therapeutic and cytotoxic properties [61]. These compounds have the ability to protect against DNA damage caused by reactive oxygen species (ROS) through the inhibition of the tyrosinase enzyme, a property which may be useful for the treatment of and prevention from ROS-related diseases [55]. 

All passion fruit germplasm had higher content (>0.40 mg catechin g^−1^) than the 0.30 mg catechin g^−1^ that was reported by Barbosa et al. in yellow passion fruit [20]; only Criollo POR1 had similar catechin content. In addition, Carmona-Hernández et al. reported a flavonoid content of 0.29 and 028 mg catechin g^−1^ for *P. edulis* f. *flavicarpa* and *P. edulis* f. *edulis*, respectively [62]; these values are lower than those obtained for the same species in this study. Criollo PICH1 is a local cultivar that can be considered for breeding because it showed the highest amount of flavonoids; consequently, this local germplasm is a potential source of these compounds.

Most of the information about polyphenols and flavonoids has been reported for yellow and purple passion fruit; for this reason, the results of this research contribute to the knowledge of these compounds in other species of *Passiflora*.

The provitamin A carotenoid (β carotene) is an important source of vitamin A [63] from fruit consumption. Passion fruit is a good source of β carotene [64]; however, the carotenoid content in passion fruit pulp depends on several factors such as the crop production system, climatic conditions and fruit maturation [65]. Carotenoids can inhibit cell oxidation processes [27] and these compounds may prevent cancer and degenerative diseases, as well as stimulate the immune system and metabolism [66]. 

The results for carotenoids obtained for all the passion fruit germplasm grown in Ecuador were higher (from 22.80 to 67.73 µg β carotene g^−1^) than those found by Wondracek et al. (7.80 µg β carotene g^−1^) [67] and Barbosa et al. (1.87 µg β carotene g^−1^) [20], but lower than the 77.00 µg β carotene g^−1^ reported by Pertizatti et al. [65]; both cases refer to yellow passion fruit grown in different places in Brazil. However, the result obtained for *P. edulis* f. *flavicarpa* (INIAP 2009 and P10) was similar to that (38.29 µg β carotene g^−1^) reported by Septembre-Malaterre et al. [60]. On the other hand, carotenoid contents of 13.34 and 13.62 µg β carotene g^−1^ have been reported for yellow passion fruit and 1.72 µg β carotene g^−1^ for purple passion fruit [1,59], values much lower than found in this study. In addition, Souza et al. found a higher content of β carotene in fruit of *P. alata* (82.49 µg g^−1^) than detected in our study (22.80 µg β carotene g^−1^) [68]. Criollo PICH1 also showed the highest amount of carotenoids, corroborating the importance of characterizing local germplasms.

The main role of vitamin C (ascorbic acid) is as an antioxidant and cofactor in redox reactions, and fruits are a good source of this compound [69]. Vitamin C has exhibited mechanisms of action against respiratory infections, which include antioxidant, anti-inflammatory, antithrombotic, and immuno-modulatory functions; moreover, preliminary studies have suggested that it might improve outcomes in the treatment of COVID-19 [70].

Passion fruit is a tropical fruit rich in Vitamin C [64]. The content of this vitamin in several *Passiflora* species varies from 2.3 to 57.7 mg 100 g^−1^ [65]. Barbosa et al. reported a vitamin C content of 26.42 mg 100 g^−1^ in yellow passion fruit [20], which is similar to values obtained for INIAP 2009, P10 (both *P. edulis* f. *flavicarpa*) and Criollo POR1 **(***Passiflora* sp.). In addition, all passion fruit germplasm, except for *P. alata*, had higher vitamin C content (>23.00 mg 100 g^−1^) than reported by Barbosa de Oliveira et al. (17.12 to 21.81 mg 100 g^−1^) [18] and Carvajal et al. (18 mg 100 g^−1^) [71]. However, Gulupa (purple passion fruit) had a higher vitamin C content than the yellow passion fruit germplasms, in this study; therefore, this germplasm could be considered for breeding purposes. However, its fruit shell is purple color, and crossing with the yellow germplasms would generate phenotypic variability in the progeny. In addition, Gulupa can be used in food industry because of its high content of vitamin C.

Antioxidant activity depends not only on the phenolic composition, but also on the presence of other bioactive compounds, their interactions and synergic effects [72]. ABTS and FRAP values showed that *P. edulis* f. *flavicarpa* germplasm had the best antioxidant activity. Barbosa de Oliveira et al. reported a total antioxidant activity of 7.20 μmol TE g^−1^ (fresh weight) for ripe yellow passion fruit grown under conventional management [18]. This study estimated, based on dry weight, an average value of 41.40 µmol TE g^−1^ using the ABTS method for *P. edulis* f. *flavicarpa,* which is higher than that reported by Carvajal et al. (27.78 µmol TE g^−1^) [71]; while *P. edulis* f. *edulis* had a value which was relatively close to the 42.34 µmol TE g^−1^ reported by the same author. Ramos et al. found that antioxidant activity was related to carotenoid content [1]. This was not observed in our study because it was correlated with the polyphenol content, corroborating that phenolic compounds are the main antioxidants, strongly contributing to the total antioxidant activity [18].

#### 4.2.2. Sugar Content and Organic Acids

Sugars and organic acids are fruit quality factors appreciated by consumers and the food industry [73]; their content can vary depending on factors such as species, cultivars, agronomic and climatic conditions. Sugars do not depend on whether the fruit is acidic or not [74]. 

It has been reported that temperature can influence total sugar content in passion fruit [75], which was reflected in the results of the yellow and purple passion fruit that are grown in different altitudes and temperature conditions in Ecuador. Studies have found a range from 5.25 to 6.98 g/100 g^−1^ of total sugars in yellow passion fruit [23,76,77]; however, Ramaiya et al. reported total sugar of 5.75 and 13.97 g 100 g^−1^ for *P. edulis* f. *flavicarpa* harvested in two different places in Malaysia [23]; these values are lower than detected in our study under Ecuadorian conditions, thus corroborating that environmental conditions influence this parameter [74]. Ramaiya et al. also found an amount of 14.28 g 100 g^−1^ for *P. edulis* f. *edulis* [23], which was lower than that detected in our study. Mamede et al. found a total sugar content of 15.40 g 100 g^−1^ of pulp of Sweet passion fruit (*P. alata*) [31], which is much lower than that found in this research (52.47 g 100 g^−1^). Sweet passion fruit and Criollo PICH1 showed the highest amount of total sugars; both are underutilized cultivars that could be exploited for this fruit trait.

Sucrose, glucose and fructose are the predominant sugars in passion fruit but their sugar content differs among species [24]. Sucrose (non-reducing sugar) is in less proportion than glucose and fructose (reducing sugars) in passion fruit [23], a trend that was also observed in this study. Sweet passion fruit had the lowest content of non-reducing sugars, with less sucrose in comparison to the other germplasms. Glucose has been found to be the main sugar in yellow passion fruit [20,23], whereas this sugar and fructose are present in similar amounts in red passion fruit [23]. 

Organic acids have antibacterial effects [78]. The main organic acids quantified in the passion fruit pulp are citric and malic acids [20]. Citric acid is a safe organic acid responsible for fruit flavor and can be used as a food additive [79]. In fruit, it inhibits bacterial and fungal growth, improves disease resistance [80] and reduces postharvest respiration [81]. These organic acids are influenced by temperature [75]. In terms of citric acid, the results for *P. edulis* f. *flavicarpa* (INIAP 2009 and P10) were higher than those (3.87 g 100 g^−1^) reported by Barbosa et al. [20]. Criollo POR1 also had a high value, thus these three genotypes were high in acidity. Malic acid is another key flavor component of fruit [82] and also used in the food industry [78]. Barbosa et al. reported low malic acid content (1.60 g 100 g^−1^) for the yellow passion fruit compared with the value obtained in this study [20]. Most fruit had citric acid levels ranging from 0.03 to 5.15 g 100 g^−1^ and malic acid levels ranging from 0.01 to 2.18 g 100 g^−1^ [83]; however, the values obtained in the passion fruit germplasm grown under Ecuadorian field conditions were higher, except for *P. alata* which was in the mentioned range for citric acid content. The breeding germplasm P10 showed a high amount of both organic acids: this could be a distinctive feature for this cultivar if it is released as a variety in the future, with potential use focused on the food industry.

### 4.3. Correlation Analysis

In this study, there was a positive correlation (Pearson) between polyphenols and ABTS and FRAP (antioxidant activity), which is in agreement with Rotta et al., who associated total phenolic content of passion fruit pulp with ABTS analysis [84]. In addition, there was also a positive partial correlation between polyphenols and FRAP which corroborates that polyphenols are the main compound related to passion fruit antioxidant activity [57]. Polyphenols and carotenoids had a negative correlation, both have been associated with antioxidant activity and ease prevention [85]. There was a negative correlation also between reducing sugars and citric acid, both compounds that are used for fruit growth and respiration [86].

Partial correlation coefficients are considered more appropriate for selection of a trait in a breeding program when considering the effect of the other variables [87]; this is because the Pearson correlation coefficient only indicates the linear correlation among two sets of data, thus indicating a trend; it does not involve all the other variables in the response. Vitamin C showed a correlation with carotenoids and malic acid, which means that if a breeding population is developed based on vitamin C the other two parameters will also be linked to the former.

### 4.4. Multivariate Analysis

One way to discriminate between samples in relation to different compounds is by a multivariate analysis using PCA [88]. According to this analysis, germplasm showing more polyphenol content had less carotenoid content; however, Ramos et al. did not find an inverse relationship between polyphenols and carotenoids using this type of analysis [1]. In addition, fruit showing more vitamin C content had less malic acid content which agrees with Ponder et al. [89] but contrasts with the results reported by Aubert and Chalot [90] in other fruit crops; this response could vary according to the fruit species. Antioxidant activity was opposite to total sugars, which is in accordance with the reports of Ramaiya et al. for passion fruit [23]. Citric acid was also inversely related to the total sugar content; most passion fruit germplasms are slightly acid because of the high citric acid content [20], which means they contain less sugar content.

PCA constitutes an efficient technique to distinguish the passion fruit species in relation to the different fruit compounds, even when the differences in the values of the traits measured are not great [20]. It was clear that yellow passion fruit (*P. edulis* f. *flavicarpa*) and Sweet passion fruit (*P. alata*) had good polyphenol and malic acid content, whereas Gulupa (*P. edulis* f. *edulis*) and the Criollos (*Passiflora* sp.) had high flavonoid, carotenoid and vitamin C contents.

Results of both analysis (PCA and k-means) were similar, showing the same trend for relevant traits such as polyphenol and carotenoid content, antioxidant activity and total sugars. In both methods, INIAP 2009, P10 (both *P. edulis* f. *flavicarpa*) and Sweet passion fruit (*P. alata*) had high polyphenol content, and the latter also showed the lowest carotenoid content. In group formation by the k-means method, INIAP 2009 and P10 were found in the same group, which make senses because P10 is a breeding germplasm that comes from INIAP 2009 segregation. This pattern can also be seen in the PCA analysis. INIAP 2009 and P10 were related to high antioxidant activity in both methods. On the other hand, the local germplasms (POR 1 and PICH1) were grouped individually, although phenotypically they would be considered as yellow passion fruit, this result means that they had different chemical characteristics from the other yellow passion fruit germplasm (INIAP 2009 and P10) and between them. 

This research has generated information on the content of biocompounds and antioxidant activity of commercial cultivars (yellow and purple passion fruit), but has also considered local germplasms that despite having a smaller fruit size and yield, showed themselves to be a source of antioxidant compounds, especially flavonoids and carotenoids (PICH1). In addition, the results showed that *P. alata* was a good source of polyphenols and reducing sugars, aspects that could be related to its medicinal properties. A limitation of this study was that the evaluated germplasms are cultivated in different environmental conditions (altitude, precipitation and heliophany) in Ecuador due to adaptations for their growth and production. Therefore, the effect of the environment is a factor which would influence the content of biocompounds. Nevertheless, this study provides information that is very useful as a reference for further chemical profile studies, breeding programs and genotype X environment interaction research in passion fruit species.

## 5. Conclusions

There were differences in the contents of antioxidant compounds, sugars and organic acids in the passion fruit germplasm assessed in this study. There was an inverse relationship between polyphenols and carotenoids, and total sugars and antioxidant activity. INIAP 2009 and P10 showed the high amount of polyphenols, antioxidant activity and citric content, Sweet passion fruit had the lowest vitamin C content while Gulupa showed the highest content. In terms of the local germplasm, POR1 showed the lowest amount of flavonoids while PICH1 had high flavonoids and carotenoid content. It was determined that polyphenols were the main compound that influenced antioxidant activity in the passion fruit species. 

These results contribute to the information on fruit phytochemical content and add value to passion fruit species as a nutraceutical source. In addition, this information can also be used to develop objectives in breeding programs for this tropical fruit. 

## Figures and Tables

**Figure 1 plants-11-00328-f001:**
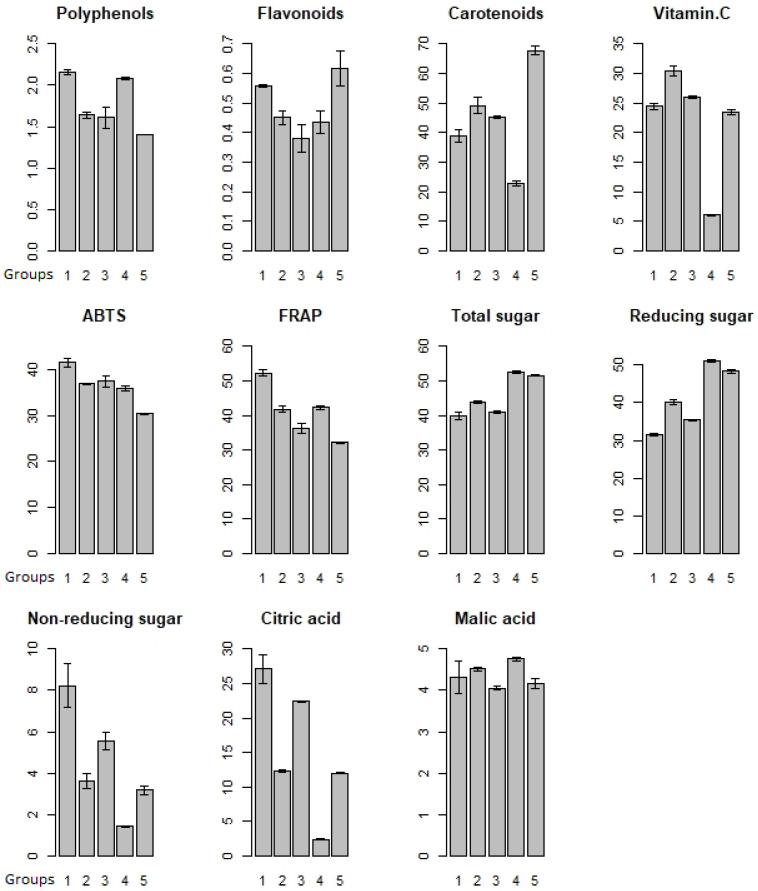
Groups (5) formed by the k-means algorithm and the elbow method showing the phytochemical traits. Group 1 *P. edulis* f. *flavicarpa* (INIAP 2009 and P10), group 2 *P. edulis* f. *edulis* (Gulupa), group 3 *Passiflora* sp. (Criollo POR1), group 4 *P. alata* (Sweet passion fruit), and group 5 *Passiflora* sp. (Criollo PICH1). Results are expressed in mg GAE g^−1^ for polyphenols, mg catechin g^−1^ for flavonoids, µg β carotene g^−1^ for carotenoids, mg 100 g pulp^−1^ for vitamin C, µmol TE g^−1^ for ABTS and FRAP, g 100 g^−1^ for sugar and organic acid content.

**Figure 2 plants-11-00328-f002:**
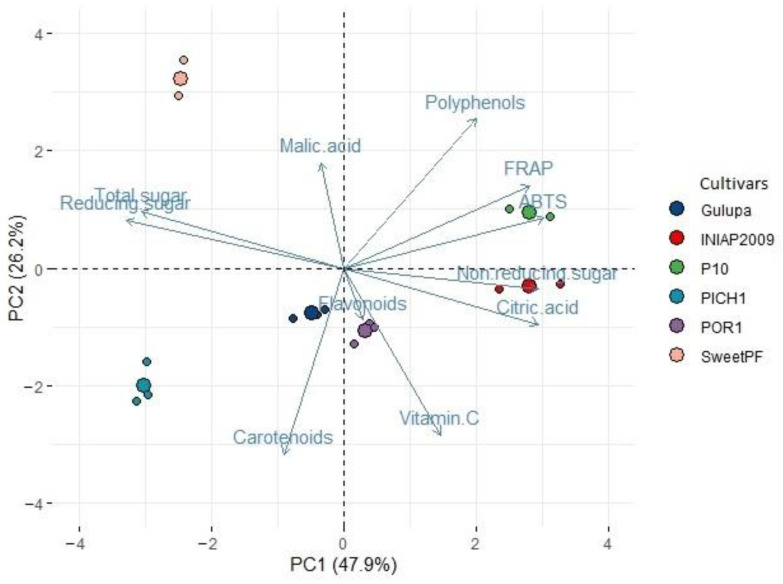
Principal component (PC) analysis of phytochemical traits of passion fruit germplasm. *Passiflora edulis* f. *flavicarpa* (INIAP 2009 and P10), *Passiflora edulis* f. *edulis* (Gulupa), *Passiflora alata* (SweetPF), *Passiflora* sp.—local germplasm (POR1 and PICH1).

**Table 1 plants-11-00328-t001:** Passiflora species analyzed to determine the fruit mineral content.

Species	Name	Type of Germplasm ^1^	Site	Province	Latitude (South)	Longitude (West)	Altitude (masl)	Annual Precipitation (mm)	Annual Average Temperature (°C)	Heliophany (Hours/Year)
*Passiflora edulis* f. *flavicarpa*	INIAP 2009	EV	Portoviejo	Manabí	01°09′43″	80°23′06″	52	852	26	1385
*Passiflora edulis* f. *flavicarpa*	P10	BG								
*Passiflora* sp.	Criollo POR1	ELG								
*Passiflora* sp.	Sweet PF	IG	Quevedo	Los Ríos	01°04′24″	79°29′14″	74	1200	25	920
*Passiflora edulis* f. *edulis*	Criollo PICH1	ELG								
*Passiflora alata*	Gulupa	IG	Tumbaco	Pichincha	00°12′57″	78°24′43″	2348	892	17	2039

^1^ EV = Ecuadorian variety, BG = Breeding germplasm, ELG = Ecuadorian local germplasm, IG = Introduced germplasm.

**Table 2 plants-11-00328-t002:** Color coordinates (L* a* b*) of pulp passion fruit from germplasm grown in Ecuador.

Germplasm	Fruit	Lyophilized Pulp	a* (+Red, −Green)	b* (+Yellow, −Blue)
INIAP 2009	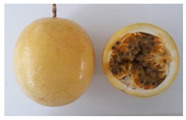	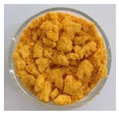	4.54 ± 0.48 c	41.15 ± 0.55 b
P10	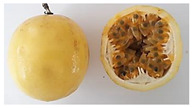	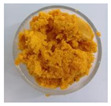	2.35 ± 0.06 d	47.56 ± 1.14 a
Sweet passion fruit	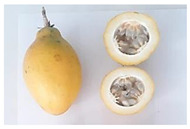	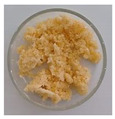	−0.46 ± 0.02 e	13.21 ± 0.17 d
Gulupa	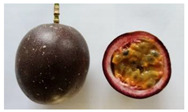	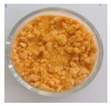	6.44 ± 0.20 b	36.17 ± 0.25 c
Criollo POR1	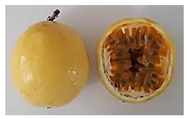	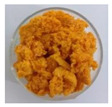	3.53 ± 0.03 c	38.00 ± 1.19 b
Criollo PICH1	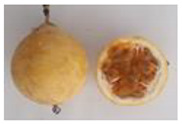	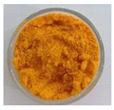	9.67 ± 0.20 a	46.75 ± 0.24 a

Different letters indicate significant differences (*p* < 0.05) using the ANOVA one-way analysis followed by Tukey’s test.

**Table 3 plants-11-00328-t003:** Lightness, chromaticity and Hue angle determined on pulp passion fruit of germplasm grown in Ecuador.

Germplasm	L* Lightness	C* Chroma	H° Hue
INIAP 2009	44.02 ± 0.48 ab	41.41 ± 0.51 b	1.46 ± 0.01 b
P10	42.98 ± 0.60 b	47.62 ± 1.13 a	1.52 ± 0.01 a
Sweet passion fruit	45.37 ± 0.13 a	13.21 ± 0.17 d	−1.54 ± 0.01 d
Gulupa	45.59 ± 0.35 a	36.74 ± 0.21 c	1.39 ± 0.01 c
Criollo POR1	38.25 ± 0.22 c	38.16 ± 1.18 bc	1.48 ± 0.01 b
Criollo PICH1	44.29 ± 0.23 ab	47.73 ± 1.13 a	1.37 ± 0.01 c

Different letters indicate significant differences (*p* < 0.05) using the ANOVA one-way analysis followed by Tukey’s test.

**Table 4 plants-11-00328-t004:** Antioxidant activity and compounds in passion fruit germplasm grown in Ecuador.

Germplasm	Polyphenols * (mg GAE g^−1^)	Flavonoids * (mg catechin g^−1^)	Carotenoids * (µg β carotene g^−1^)	Vitamin C ** (mg 100 g pulp^−1^)	ABTS * (µmol TE g^−1^)	FRAP * (µmol TE g^−1^)
INIAP 2009	2.09 ± 0.01 a	0.55 ± 0.01 ab	42.51 ± 0.31 bc	25.57 ± 0.58 bc	43.00 ± 0.93 a	53.61 ± 1.16 a
P10	2.22 ± 0.03 a	0.56 ± 0.01 ab	35.20 ± 2.19 c	23.24 ± 0.38 c	39.80 ± 0.70 ab	50.71 ± 0.89 a
Sweet passion fruit	2.08 ± 0.01 a	0.44 ± 0.04 bc	22.80 ± 0.95 d	6.07 ± 0.12 d	35.84 ± 0.43 c	42.18 ± 0.51 b
Gulupa	1.64 ± 0.03 b	0.45 ± 0.02 abc	49.19 ± 2.58 b	30.44 ± 0.89 a	36.80 ± 0.03 bc	41.69 ± 0.86 b
Criollo POR1	1.61 ± 0.12 b	0.38 ± 0.05 c	45.37 ± 0.44 b	25.96 ± 0.22 b	37.35 ± 1.20 bc	36.27 ± 1.52 c
Criollo PICH1	1.41 ± 0.01 b	0.62 ± 0.06 a	67.73 ± 1.37 a	23.52 ± 0.44 c	30.32 ± 0.06 d	32.14 ± 0.27 c

Different letters indicate significant differences (*p* < 0.05) using the ANOVA one-way analysis followed by Tukey’s test. * dry weight basis, ** fresh weight basis.

**Table 5 plants-11-00328-t005:** Sugar composition of passion fruit germplasm grown in Ecuador.

Sugar Composition	Total Sugar (g 100 g^−1^)	Reducing Sugar (g 100 g^−1^)	Non-Reducing Sugars (g 100 g^−1^)	Citric Acid (g 100 g^−1^)	Malic Acid (g 100 g^−1^)
INIAP 2009	37.77 ± 0.74 d	31.20 ± 0.31 e	6.57 ± 1.05 ab	22.50 ± 0.01 b	3.43 ± 0.02 d
P10	41.83 ± 0.77 bc	31.94 ± 0.51 e	9.90 ± 1.28 a	31.77 ± 0.25 a	5.19 ± 0.04 a
Sweet passion fruit	52.47 ± 0.36 a	51.03 ± 0.34 a	1.43 ± 0.02 c	2.46 ± 0.02 d	4.75 ± 0.05 b
Gulupa	43.86 ± 0.33 b	40.25 ± 0.69 c	3.61 ± 0.36 bc	12.29 ± 0.17 c	4.51 ± 0.04 b
Criollo POR1	41.01 ± 0.31 c	35.44 ± 0.11 d	5.56 ± 0.41 b	22.41 ± 0.10 b	4.05 ± 0.05 c
Criollo PICH1	51.47 ± 0.23 a	48.27 ± 0.44 b	3.19 ± 0.21 bc	12.03 ± 0.07 c	4.16 ± 0.13 c

Different letters indicate significant differences (*p* < 0.05) using the ANOVA one-way analysis followed by Tukey’s test. All parameters are in dry weight basis.

**Table 6 plants-11-00328-t006:** Pearson correlation (upper triangle) and partial correlation (lower triangle) for the relationship between the contents of antioxidant compounds, sugars and organic acids in passion fruit germplasm grown in Ecuador.

	Vitamin C	Polyphenols	Flavonoids	ABTS	FRAP	Carotenoids	Total Sugars	Reducing Sugars	Non-Reducing Sugars	Citric Acid	Malic Acid
Vitamin C	-	−0.40	0.12	0.18	0.03	0.63 *	−0.64 *	−0.62 *	0.42	0.56	−0.34
Polyphenols	−0.40	-	0.12	0.65 *	0.85 **	−0.79 **	−0.27	−0.36	0.43	0.30	0.29
Flavonoids	0.05	0.37	-	−0.16	0.16	0.42	0.08	−0.03	0.24	0.21	−0.08
ABTS	−0.09	−0.34	−0.29	-	0.87	−0.50	−0.78 **	−0.79 **	0.61	0.56	−0.19
FRAP	0.26	0.62 *	0.39	0.68 **	-	−0.54	−0.58	−0.65 *	0.61	0.48	0.02
Carotenoids	0.62 *	−0.35	0.42	−0.26	0.06	-	0.06	0.07	−0.06	0.04	−0.40
Total sugars	−0.09	−0.15	−0.31	−0.19	0.31	0.06	-	0.96 ***	−0.65	−0.78 **	0.37
Reducing sugars	0.09	0.15	0.31	0.19	−0.31	−0.06	1.00 ***	-	−0.84 **	−0.90 ***	0.21
Non-reducing sugars	0.09	0.15	0.31	0.19	−0.31	−0.06	1.00 ***	−1.00 ***	-	0.91 ***	0.14
Citric acid	−0.41	0.01	0.36	−0.03	−0.35	0.22	0.05	−0.05	−0.05	-	0.01
Malic acid	0.71 **	0.10	−0.30	−0.37	0.18	−0.63	−0.26	0.26	0.26	0.46	-

* Represents statistical significant coefficients at 10%, ** at 5% and *** at 1%.

## Data Availability

Data is contained within the article.

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
