# Peer review of "Phytochemical Composition and Antioxidant Activity of *Passiflora* spp. Germplasm Grown in Ecuador"

_plants, 2022, doi:10.3390/plants11030328_

Round 1

Reviewer 1 Report

This manuscript was well-written, but the author should add more information in the introduction part about the aim of this experiment and cited in recent literatures. 

Author Response

Comment: This manuscript was well-written, but the author should add more information in the introduction part about the aim of this experiment and cited in recent literatures. 

Reply: Information from recent papers and related to the aim of this research has been added to the introduction. The references list has been updated with the new references.

Reviewer 2 Report

Authors should added in the tables and graphics the error bar

Author Response

Comment: Authors should added in the tables and graphics the error bar.

Reply: Standard error has been added to all tables and error bar to the graphics.

Reviewer 3 Report

The authors are focused on the Phytochemical composition and antioxidant activity of Passiflora spp. germplasm grown in Ecuador. Extensive results part must be appreciated but the paper looks slightly untidy due to punctuation marks, incorrect insertion of references, misspelled words, missing interspaces between sections (please see the Author's Instructions which are given to be followed), etc. Please see below my suggestions regarding this manuscript:

Please remove the point after the title of the article.

Keywords must to reflect the main characteristic words of the paper (usually reflected also by the title). So, I suggest the following keywords: phytochemical composition; antioxidant activity;  Passiflora spp. germplasm; Ecuador; polyphenols; organic acids.

I have never seen such an expression in articles "[ ] has reported", "... by method of [ ]"..... References must be inserted after some statement, not before. Impersonal manner of addressing would me more proper: it was reported, it was found, literature data underlines, etc...., inserting the references at the final of the sentences/phrases. Or, instead [ ] use Name of the first author et al. found, developed, discovered, confirmed, etc, and the number of the reference must be inserted at the final of that statement. Please revise the entire manuscript in this regard.

As the last, separate paragraph of Introduction please make the aim of the study relevant. What makes special this study? Which is its novelty character or its special aspects? Why have the author chosen this topic? What differentiate this paper from others in the same topic? Actual L87-89 are not relevant at all in this regard, there are tenths/hundreds of papers in the same topic.

L139, L140 mL not ml. Please be consistent with denotation. Please change in the entire manuscript ml with mL as Litter being the international unit of measure for volume.

L239. Replace 2.15 with 2.13. Check again all numbers of the sections/subsections. 

In the Statistical analysis section please also check that all computers programs used for analysis to be provided together with their variants.

Under Figure 1 please explain all abbreviations used on the figure.

L353. It must be reshaped as: Antioxidants are present in many fruits and vegetables, their unique chemical structure imprinting both their properties and their dietary and therapeutic actions as the literature confirmed [Glevitzky I., et al. Statistical Analysis of the Relationship Between Antioxidant Activity and the Structure of Flavonoid Compounds. Rev. Chim. 2019, 70(9), 3103-3107. https://doi.org/10.37358/RC.19.9.7497 ]

L358. It must be completed that: The importance of climatic factors  and anthropogenic aspects  must not be neglected, as their influences on the fruit production is overwhelming [Bungau et al. Expatiating the impact of anthropogenic aspects and climatic factors on long term soil monitoring and management. Environ Sci. Pollut. Res. 2021, 202, 30528-30550. https://doi.org/10.1007/s11356-021-14127-7 ; Gitea, M.A. et al. Orchard management under the effects of climate change: implications for apple, plum, and almond growing. Environ Sci Pollut Res. 2019, 269908–9915. https://doi.org/10.1007/s11356-019-04214-1 ]

L357. Plants not plans.

L360. please add point after [40]. Please carefully revise the entire manuscript regarding punctuation marks.

After L499, as the last part of this section, please describe the strengths and limitations of your study.

Author Response

Comment: Please remove the point after the title of the article.

Reply: Dot has been removed.

Comment: Keywords must to reflect the main characteristic words of the paper (usually reflected also by the title). So, I suggest the following keywords: phytochemical composition; antioxidant activity; Passiflora spp. germplasm; Ecuador; polyphenols; organic acids.

Reply: Keywords have been replaced for the reviewer´s suggestion.

Comment: I have never seen such an expression in articles "[ ] has reported", "... by method of [ ]"..... References must be inserted after some statement, not before. Impersonal manner of addressing would me more proper: it was reported, it was found, literature data underlines, etc...., inserting the references at the final of the sentences/phrases. Or, instead [ ] use Name of the first author et al. found, developed, discovered, confirmed, etc, and the number of the reference must be inserted at the final of that statement. Please revise the entire manuscript in this regard.

Reply: The name of the first author et al. has been placed at the beginning of the sentence and the reference number at the end of the statement; in some cases, the author name was added before the reference number. All the manuscript was revised and corrected.

Comment: As the last, separate paragraph of Introduction please make the aim of the study relevant. What makes special this study? Which is its novelty character or its special aspects? Why have the author chosen this topic? What differentiate this paper from others in the same topic? Actual L87-89 are not relevant at all in this regard, there are tenths/hundreds of papers in the same topic.

Reply: The novelty of this study has been stated in the last paragraph of the introduction.

Comment: L139, L140 mL not ml. Please be consistent with denotation. Please change in the entire manuscript ml with mL as Litter being the international unit of measure for volume.

Reply: The abbreviation mL was placed in the whole manuscript.

Comment: L239. Replace 2.15 with 2.13. Check again all numbers of the sections/subsections.

Reply: The sections has been checked and re-numbered.

Comment: In the Statistical analysis section please also check that all computers programs used for analysis to be provided together with their variants.

Reply: The standard error has been added to the data in all the tables and error bar in the Figure 2.

Comment: Under Figure 1 please explain all abbreviations used on the figure.

Reply: All abbreviations have been explained in the title of Figure 1.

Comment: L353. It must be reshaped as: Antioxidants are present in many fruits and vegetables, their unique chemical structure imprinting both their properties and their dietary and therapeutic actions as the literature confirmed [Glevitzky I., et al. Statistical Analysis of the Relationship Between Antioxidant Activity and the Structure of Flavonoid Compounds. Rev. Chim. 2019, 70(9), 3103-3107. https://doi.org/10.37358/RC.19.9.7497 ]

Reply: The text and the reference has been added.

Comment: L358. It must be completed that: The importance of climatic factors and anthropogenic aspects  must not be neglected, as their influences on the fruit production is overwhelming [Bungau et al. Expatiating the impact of anthropogenic aspects and climatic factors on long term soil monitoring and management. Environ Sci. Pollut. Res. 2021, 202, 30528-30550. https://doi.org/10.1007/s11356-021-14127-7 ; Gitea, M.A. et al. Orchard management under the effects of climate change: implications for apple, plum, and almond growing. Environ Sci Pollut Res. 2019, 26, 9908–9915. https://doi.org/10.1007/s11356-019-04214-1 ]

Reply: The text and the reference has been added.

Comment: L357. Plants not plans.

Replay: The word “plants” has been corrected.

Comment: L360. please add point after [40]. Please carefully revise the entire manuscript regarding punctuation marks.

Reply: The dot has been added at the end. The whole manuscript was checked.

Comment: After L499, as the last part of this section, please describe the strengths and limitations of your study.

Reply: A paragraph about the strengths and limitations of the study was added at the end of the discussion section.

Reviewer 4 Report

The manuscript entitled "Phytochemical composition and antioxidant activity of Passiflora spp. germplasm grown in Ecuador" by Viera et al., is a screening article that presents the different phytochemical compositions of various passion fruit cultivars, located in Ecuador.
This work focuses only upon 6 varieties. Not all of them are cultivated in the same area. The different microclimate could be a factor that alters the various phytochemical compounds of each variety. For safer results, all of them should be located in the same field, with the same soil, microclimate conditions, to focus upon the cultivar's genetic potential to produce the various compounds.
In the introduction section, many references that have to do with the different reported content of the antioxidant activity, carotenoids, and sugars, could be included within the discussion section.
PCA analysis should be at the end of the results since it provides more in-depth results of all the analyzed compounds.
The discussion section needs more scientific justifications since in most cases the results are only presented and are compared with those found from other research groups.
In its current form, the work presents data that have limited potential to contribute safely to the knowledge regarding the ability of those cultivars to exert their full potential towards the production of specific phytochemicals.
The text must be checked by a native English Speaker since there are multiple word mistakes.

Author Response

Comment: The manuscript entitled "Phytochemical composition and antioxidant activity of Passiflora spp. germplasm grown in Ecuador" by Viera et al., is a screening article that presents the different phytochemical compositions of various passion fruit cultivars, located in Ecuador.
This work focuses only upon 6 varieties. Not all of them are cultivated in the same area. The different microclimate could be a factor that alters the various phytochemical compounds of each variety. For safer results, all of them should be located in the same field, with the same soil, microclimate conditions, to focus upon the cultivar's genetic potential to produce the various compounds.

Reply: At the end of the discussion, the factor that the passion fruit germplasms are grown in different environmental conditions (locations) in Ecuador has been pointed like a limitation of the study and mentioned that it influenced the phytochemical composition; but also it has been mentioned that the environmental conditions where they are cultivated are the best for their grown and production. Moreover, in the case of the purple passion fruit (Gulupa) is not possible to grow it in the same conditions that the yellow passion fruit because it needs less temperature (thus more altitude) for flowering and fruiting. It has been mentioned that these results can be used as reference for further research. 

Comment: In the introduction section, many references that have to do with the different reported content of the antioxidant activity, carotenoids, and sugars, could be included within the discussion section.

Reply: The references about antioxidant activity, carotenoids and sugars have been included in the discussion.

Comment: PCA analysis should be at the end of the results since it provides more in-depth results of all the analyzed compounds.

Reply: The whole section of the results has been changed, started from fruit color, univariate analysis, correlations and PCA analysis was changed to the end of the results. The discussion of this results was also reorganized.

Comment: The discussion section needs more scientific justifications since in most cases the results are only presented and are compared with those found from other research groups. In its current form, the work presents data that have limited potential to contribute safely to the knowledge regarding the ability of those cultivars to exert their full potential towards the production of specific phytochemicals.

Reply: The discussion has been improved according to the suggestion, adding more scientific justifications, and it also has been reorganized.

Comment: The text must be checked by a native English Speaker since there are multiple word mistakes.

Reply: The manuscript was checked by Dr. Randy Kutcher for the University of Saskatchewan who is native speaker from Canada.

Round 2

Reviewer 3 Report

The authors responded to all my requests.

Reviewer 4 Report

The manuscript is accepted for publication in its present form